# *Salmonella enterica* Serovar Typhi Induces Host Metabolic Reprogramming to Increase Glucose Availability for Intracellular Replication

**DOI:** 10.3390/ijms221810003

**Published:** 2021-09-16

**Authors:** Jingting Wang, Shuai Ma, Wanwu Li, Xinyue Wang, Di Huang, Lingyan Jiang, Lu Feng

**Affiliations:** 1The Key Laboratory of Molecular Microbiology and Technology, Ministry of Education, Nankai University, Tianjin 300457, China; jingtingwang@mail.nankai.edu.cn (J.W.); 2120191113@mail.nankai.edu.cn (S.M.); 1120180080@mail.nankai.edu.cn (W.L.); wangxinyue1120200566@mail.nankai.edu.cn (X.W.); huangdi@nankai.edu.cn (D.H.); 2Tianjin Key Laboratory of Microbial Functional Genomics, TEDA Institute of Biological Sciences and Biotechnology, Nankai University, Tianjin 300457, China

**Keywords:** *S.* Typhi, glycolysis, glucose utilization, intracellular replication, pathogenesis

## Abstract

*Salmonella enterica* serovar Typhi (*S.* Typhi) is a human-limited intracellular pathogen and the cause of typhoid fever, a severe systemic disease. Pathogen–host interaction at the metabolic level affects the pathogenicity of intracellular pathogens, but it remains unclear how *S.* Typhi infection influences host metabolism for its own benefit. Herein, using metabolomics and transcriptomics analyses, combined with in vitro and in vivo infection assays, we investigated metabolic responses in human macrophages during *S*. Typhi infection, and the impact of these responses on *S.* Typhi intracellular replication and systemic pathogenicity. We observed increased glucose content, higher rates of glucose uptake and glycolysis, and decreased oxidative phosphorylation in *S.* Typhi-infected human primary macrophages. Replication in human macrophages and the bacterial burden in systemic organs of humanized mice were reduced by either the inhibition of host glucose uptake or a mutation of the bacterial glucose uptake system, indicating that *S.* Typhi utilizes host-derived glucose to enhance intracellular replication and virulence. Thus, *S.* Typhi promotes its pathogenicity by inducing metabolic changes in host macrophages and utilizing the glucose that subsequently accumulates as a nutrient for intracellular replication. Our findings provide the first metabolic signature of *S.* Typhi-infected host cells and identifies a new strategy utilized by *S.* Typhi for intracellular replication.

## 1. Introduction

*Salmonella* is a genus of facultative anaerobic Gram-negative bacteria comprising two species, *S. enterica* and *S. bongori* [1]. Although more than 2600 *Salmonella* serovars have been identified, most human infections are caused by a limited number of serovars, such as *S. enterica* serovar Typhi (*S.* Typhi), the cause of typhoid fever, a severe, life-threatening systemic disease. Globally, there are an estimated 12 to 27 million cases of typhoid fever and 200,000 typhoid fever-related deaths per year [2,3]. Transmission of *S.* Typhi usually occurs via contaminated food or water [4]. The ability of *S*. Typhi to replicate inside human macrophages is key to its pathogenicity; it is spread systemically via infected macrophages, resulting in typhoid fever [5].

Unlike most *S. enterica* serovars, which have a broad range of hosts and generally cause self-limiting gastroenteritis, *S.* Typhi is host-restricted to humans [6]. In the absence of a suitable animal model, the pathogenesis and virulence factors of *S.* Typhi are poorly understood; much of what is known has been extrapolated from studies on *S. enterica* serovar Typhimurium (*S.* Typhimurium), which causes a systemic disease resembling human typhoid fever in mice [7]. However, the pathological mechanisms of diseases caused by *S*. Typhi and *S*. Typhimurium are markedly different [8]. Notably, *Salmonella* pathogenicity island (SPI)-2 genes, present in both *S.* Typhi and *S.* Typhimurium, are essential for *S*. Typhimurium replication inside murine macrophages but are not required for *S*. Typhi replication in human macrophages [9,10]. SPI-7, which encodes a Vi capsular antigen specific to *S*. Typhi, while absent from the genome of *S.* Typhimurium, is known to contribute to *S*. Typhi systemic infection [11,12]. Thus, *S*. Typhimurium infection in mouse models cannot fully mimic *S*. Typhi pathogenicity in human typhoid fever. However, humanized mice have recently emerged as an essential tool in the study of *S.* Typhi pathogenicity. Using transposon-directed insertion site sequencing and humanized mice, Karlinsey et al. revealed that biosynthesis of the Vi capsule, lipopolysaccharide, aromatic amino acids, and the siderophore, salmochelin, were essential for *S.* Typhi virulence in humanized mice, whereas the PhoPQ two-component system, SPI-2 genes, and CdtB typhoid toxin were not required for lethal *S.* Typhi virulence [2,13,14,15].

The interaction of intracellular bacterial pathogens with host cells at the metabolic level is critical to their pathogenicity. Bacterial infection initiates a host antibacterial response that is partially dependent on the alterations in host glucose metabolism, which is known as a Warburg-like metabolism. The Warburg metabolism is the most common metabolic change in activated host immune cells, characterized by increased glucose uptake and glycolysis and decreased oxidative phosphorylation (OXPHOS) via the tricarboxylic acid (TCA) cycle [16]. Warburg-like metabolism is essential for host cells to combat invading microbes by producing inflammatory cytokines, reactive oxygen and nitrogen species, prostaglandins, and itaconate [17]. However, several intracellular pathogens, including *Legionella pneumophila*, *Chlamydia trachomatis*, and *Francisella tularensis*, can modulate host glucose metabolism to enhance their own virulence, mainly by promoting or inhibiting Warburg-like metabolism [18,19,20,21]. These modulations increase the availability of nutrients, activate virulence factors, or inhibit host immune responses in favor of intracellular bacterial replication. We recently showed that *S*. Typhimurium infection of murine macrophages also induces a Warburg-like metabolism. Moreover, *S.* Typhimurium suppressed serine synthesis in the host via the type III secretion system (T3SS) effector, SopE2, promoting the accumulation of glycolytic intermediates, especially 3-phosphoglycerate (3PG), pyruvate, and lactate [22]. Of these, *S.* Typhimurium used 3PG as an intracellular growth nutrient, whereas pyruvate and lactate were used as signals to induce SPI-2 gene expression [22]. Although *S.* Typhi and *S.* Typhimurium share many virulence factors, they also exhibit distinct features in association with specific hosts. Thus, it is possible that *S.* Typhi infection also induces host metabolic changes, either similar to or different from *S*. Typhimurium infection, to promote its virulence.

In this study, using combined metabolomics, transcriptomics, and in vitro and in vivo infection assays, we investigated whether *S*. Typhi infection induces metabolic changes in human macrophages and, if so, how the host metabolic changes influence *S.* Typhi intracellular survival and pathogenicity. Our findings suggest that S. Typhi manipulates host cell metabolism and utilizes host-derived glucose as a nutrient for intracellular replication and systemic virulence.

## 2. Results

### 2.1. S. Typhi Infection Increases Glycolysis and Decreases OXPHOS in Human Macrophages

To investigate whether *S.* Typhi infection affects host metabolism, monocyte-derived macrophages (MDMs) were infected or mock-infected with *S.* Typhi for 8 h, followed by measurements of the extracellular acidification rate (ECAR) and oxygen consumption rate (OCR). ECAR reflects glycolytic flux and is primarily a measure of lactate production, whereas OCR is an indicator of mitochondrial respiration, i.e., OXPHOS [23]. Compared with mock-infected MDMs, *S.* Typhi-infected MDMs showed an increased ECAR and a decreased OCR (Figure 1A,B). Consequently, the ratio between mitochondrial respiration and glycolysis (OCR/ECAR ratio) was significantly decreased (Figure 1C). The results suggest that *S.* Typhi infection promoted host glycolytic flux while reducing OXPHOS.

Transcriptomics and targeted metabolomics analyses were conducted to characterize the metabolic changes of MDMs following *S.* Typhi infection. Expression of the major glucose transporter, solute carrier family 2 facilitated glucose transporter member 1 (SLC2A1), was higher in *S*. Typhi-infected MDMs (5.53-fold) than in mock-infected MDMs (Figure 2A), suggesting enhanced glucose uptake in MDMs following *S.* Typhi infection. While gene expression of lactate dehydrogenase A (*LDHA*) was moderately increased (1.44-fold), in agreement with increased ECAR (Figure 1A), gene expression of other glycolytic enzymes were either moderately increased (bisphosphoglycerate mutase: *BPGM*, 2.66-fold; phosphofructokinase: *PFKP*, 1.43-fold; phosphoglycerate mutase 1: *PGAM1*, 1.34-fold), moderately decreased (hexokinase, *HK*: 1.59-fold; glucose-6-phosphate isomerase: *GPI*, 1.54-fold), or unchanged (Figure 2A). However, the expression of *PFKFB3*, encoding the bifunctional enzyme fructo-2-kinase/fructose-2,6-biphosphatase enzyme (PFK2), was highly increased in *S*. Typhi-infected MDMs (9.21-fold) (Figure 2A). PFK2 catalyzes the conversion of fructose 6-phosphate (F6P) to fructose-2,6-bisphosphate (F-2,6-P_2_), which is the most potent allosteric activator of phosphofructokinase-1 (PFK1), the rate-limiting enzyme in glycolysis [24]. The increased expression of *PFKFB3* results in a higher rate of glycolysis [25,26]. In accordance with the transcriptomics data, metabolomics analysis revealed that *S.* Typhi-infected MDMs had higher lactate levels, while the levels of other glycolysis intermediates remained unchanged, with the exception of glucose-6-phosphate (G6P), which showed decreased levels (Figure 2B). The lower G6P content may be the result of increased *PFKFB3* expression, which increased the consumption rate of G6P, and decreased *HK* expression, thereby slowing the rate of G6P formation. These results indicate that glycolysis is increased in *S*. Typhi-infected MDMs.

In line with the decreased mitochondrial respiration indicated by the OCR data (Figure 1B), gene expression of almost all the enzymes involved in OXPHOS and the TCA cycle was downregulated in *S.* Typhi-infected MDMs (Figure 2A). Metabolomics analysis showed increased levels of the TCA cycle intermediates, citrate, succinate, and malate, while the levels of α-ketoglutaric acid (α-KG) and fumarate were not significantly different (Figure 2B), contrary to the decreased gene expression of TCA cycle enzymes. The increase in these TCA cycle intermediates was probably a result of glutamine-dependent anerplerosis, during which glutamine is converted to glutamate, and then to α-KG, to replenish the pools of the TCA cycle and maintain mitochondrial functions [27]. The glutamine-derived α-KG is subsequently converted to succinate, fumarate, and malate but can also undergo reductive carboxylation by the NADPH-linked mitochondrial isocitrate dehydrogenase (IDH2) to form isocitrate, which can then be isomerized to citrate [28,29]. Consistent with this hypothesis, the expression of glutaminase (GLS), which catalyzes the first and rate-limiting step of glutamine-dependent anaplerosis, was increased in *S.* Typhi-infected MDMs (2.20-fold).

Collectively, these data suggest that *S.* Typhi infection of human macrophages results in increased glycolysis and downregulation of the TCA cycle and OXPHOS.

### 2.2. Increased Lactate Production in Macrophages Has No Effect on S. Typhi Intracellular Replication

Lactate can be utilized by several intracellular pathogens as a growth nutrient inside host macrophages, including *Mycobacterium tuberculosis* and *Brucella abortus* [21,30]. Considering that lactate production is increased in human macrophages following *S.* Typhi infection, we investigated whether the increase in host-derived lactate contributes to *S.* Typhi intracellular replication by constructing an *lldP* mutant strain (Δ*lldP*) that cannot utilize lactate. We found that the mutation of *lldP* did not influence the replication ability of *S*. Typhi in human THP-1 cells (Figure 3A). We subsequently tested the effect of exogenous lactate on *S.* Typhi intracellular replication by infecting THP-1 cells that were cultured with RPMI-1640 medium containing 0, 6, or 12 mM lactate during infection. The replication ability of *S.* Typhi in THP-1 cells was not affected by the addition of exogenous lactate at either concentration (Figure 3B). The results indicate that the utilization of lactate is not required for *S*. Typhi to replicate in human macrophages.

### 2.3. Increased Glucose Levels in Macrophages Contribute to S. Typhi Intracellular Replication

Transcriptomics and metabolomics analysis revealed increased expression of the major glucose transporter, SLC2A1. However, the decreased expression of *HK* and lower G6P levels in *S.* Typhi-infected MDMs suggest that the glucose uptake rate was increased while the utilization rate of glucose by glycolysis was decreased. We, therefore, hypothesized that *S.* Typhi infection increases the glucose content of human macrophages.

We first verified the increased glucose uptake in MDMs following *S.* Typhi infection. MDMs were infected or mock-infected with *S.* Typhi for 8 h, and subsequently cultured for 30 min with the fluorescent glucose analog, 2-(*N*-(7-nitrobenz-2-oxa-1,3-diazol-4-yl) amino)-2-deoxyglucose (2-NBDG). Fluorescence spectroscopy and confocal microscopy indicated that the 2-NBDG fluorescence signal was stronger in the *S.* Typhi-infected MDMs than in the mock-infected MDMs (Figure 4A,B), indicating that the uptake of the fluorescent glucose analog was increased in *S.* Typhi-infected cells. Next, we measured the glucose concentrations in *S.* Typhi-infected and mock-infected MDMs using a fluorimetric glucose assay kit. The glucose concentration was significantly higher in *S.* Typhi-infected MDMs than in mock-infected MDMs (Figure 4C). These results suggest that *S.* Typhi infection increases glucose levels in human macrophages.

Next, we investigated whether the increased glucose content of macrophages during *S.* Typhi infection was associated with *S.* Typhi intracellular replication. To reduce the intracellular glucose level, we knocked down *slc2a1* expression using lentiviral short hairpin RNA (shRNA) in THP-1 cells. We verified the decrease in intracellular glucose content in the sh-*slc2a1*-treated cells compared to that of the control shRNA-treated cells using a fluorimetric glucose assay kit (Figure 5A). We then tested the replication ability of *S.* Typhi in sh-*slc2a1*-treated cells and control shRNA-treated cells. The replication ability of *S.* Typhi in sh-*slc2a1*-treated cells was significantly lower than in the control shRNA-treated cells (Figure 5B), indicating that the decrease in macrophage glucose level reduces *S.* Typhi replication in human macrophages.

The effect of exogenous glucose on the intracellular replication of *S.* Typhi was investigated. We infected THP-1 cells and sh-*slc2a1*-treated THP-1 cells that were cultured in RPMI-1640 medium containing 0, 1, or 2 mg/mL glucose during infection. The results showed that the replication ability of *S.* Typhi in both THP-1 cells and sh-*slc2a1*- treated THP-1 cells was increased with the exogenous glucose concentration (Figure 5C,D), revealing the importance of host glucose for *S.* Typhi intracellular replication.

Together, these data indicate that the increased macrophage glucose level upon *S.* Typhi infection contributes to *S.* Typhi intracellular replication.

### 2.4. S. Typhi Uses Host-Derived Glucose as a Nutrient for Intracellular Replication to Promote Systemic Infection

Many intracellular pathogens utilize glucose for intracellular survival and growth, including *S.* Typhimurium [31]. To verify whether this is also the case for *S*. Typhi, we constructed a *ptsG-manXYZ-glk* triple mutant strain (Δ*ptsG/manXYZ/glk*) that cannot utilize glucose [32,33]. PtsG is the major glucose-specific enzyme in the permease of phosphotransferase system (PTS) superfamily in *Salmonella* [34]; *manXYZ* encodes components of the mannose phosphotransferase (Man-PTS) system, which is also a route of entry for glucose into bacterial cells [35], while *glk* encodes a glucokinase that phosphorylates glucose in bacteria [36]. We found that the replication of Δ*ptsG/manXYZ/glk* was significantly suppressed in THP-1 cells compared with the wild-type strain, indicating that glucose uptake and utilization were essential for *S.* Typhi intracellular replication (Figure 6A). The replication ability of Δ*ptsG/manXYZ/glk* was similar in sh*slc2a1*-treated THP-1 cells and control shRNA-treated THP-1 cells, indicating that the replication of the mutant was not affected by the host cell glucose content, consistent with its inability to utilize glucose (Figure 6B). Confocal fluorescence microscopy indicated that the fluorescent glucose analog, 2-NBDG, was localized near the cell nucleoli, and was co-localized with the bacteria in the wild-type-infected cells, but was dispersed throughout the cytoplasm in Δ*ptsG/manXYZ/glk*-infected cells (Figure 6C), thus confirming that the mutant was unable to utilize the host-derived glucose. Therefore, *S*. Typhi utilizes host-derived glucose as a nutrient for intracellular replication in human macrophages.

The contribution of glucose utilization to *S.* Typhi systemic virulence was investigated using in vivo competition assays in humanized mice that were infected with equal numbers of wild-type or Δ*ptsG/manXYZ/glk* via intraperitoneal administration and euthanized 72 h later. Their livers and spleens were harvested to enumerate colony-forming units (CFUs, and the competitive index was calculated as the ratio of recovered Δ*ptsG/manXYZ/glk* CFUs to wild-type CFUs, normalized to the ratio in the inoculum. The result showed that Δ*ptsG/manXYZ/glk* was significantly outcompeted by the wild-type in the spleens and livers of the humanized mice (Figure 6D). Therefore, the utilization of host-derived glucose for intracellular replication contributes significantly to *S*. Typhi systemic pathogenicity.

## 3. Discussion

Herein, we demonstrated that *S.* Typhi infection induces metabolic changes in human macrophages, resulting in glucose accumulation while promoting a Warburg-like phenotype by increasing glycolysis and decreasing OXPHOS. We also showed that *S.* Typhi uses macrophage-derived glucose as a nutrient for intracellular replication and systemic infection, indicating that the reprogramming of host metabolism to increase glucose availability is an important event during *S.* Typhi infection.

Enhanced glucose uptake and glycolysis were also detected in murine macrophages infected by the closely related serovar, *S.* Typhimurium, but different metabolic features were also observed [22,37,38]. Although lactate content is increased in both *S.* Typhimurium-infected murine macrophages and in *S.* Typhi-infected human macrophages, an increase in 3PG levels occurs only in the former, while an increase in glucose levels only occurs in the latter. 3PG, which is used by *S.* Typhimurium as a nutrient for intracellular growth, has no effect on *S.* Typhi replication, as indicated by the mutation of the 3PG transporter gene *pgtP* (Appendix A). Lactate induces SPI-2 gene expression in *S*. Typhimurium, which is essential for *S.* Typhimurium to replicate in murine macrophages. Moreover, *S.* Typhi can also respond to lactate to activate its SPI-2 genes (Appendix A). However, SPI-2 is not required for *S.* Typhi replication in human macrophages [9]. Thus, the induction of SPI-2 gene expression is not as important as in *S.* Typhimurium. Therefore, the specific metabolic changes in macrophages induced by *S.* Typhi or *S.* Typhimurium infection are probably associated with their respective growth demands as well as the virulence requirements inside respective macrophages.

*S.* Typhi infection increased glucose uptake and enhanced glycolysis in the human macrophages, which are characteristic features of the Warburg-like metabolism. However, the glucose content was also increased in *S.* Typhi-infected human macrophages, likely as a result of decreased gene expression of HK, which catalyzes the phosphorylation of glucose to G6P—the first, irreversible step in glycolysis. Consequently, only a fraction of the macrophage-derived glucose undergoes glycolysis, resulting in a Warburg-like phenotype of the human macrophages, and the rest is available for *S*. Typhi utilization. This explains our observation of moderate upregulation of glycolysis in macrophages, as indicated by the lack of increase in gene expression of most glycolytic enzymes and the production of intermediates. By contrast, the gene expression of HK was increased, along with other glycolysis-related genes, in response to increased glucose uptake in *S.* Typhimurium-infected murine macrophages. Consequently, more macrophage-derived glucose undergoes glycolysis, as indicated by decreased glucose levels and increased accumulation of glycolysis intermediates [22]. Therefore, it appears that *S.* Typhi suppresses HK gene expression in human macrophages to accumulate host glucose for its own intracellular replication, although how it accomplishes this is currently unclear and merits further investigation.

Our findings suggest that *S*. Typhi reprograms host metabolism in human macrophages to increase glucose levels and uses it as a nutrient for intracellular replication, thus elucidating a new mechanism underlying *S*. Typhi pathogenesis. The *S.* Typhi-induced metabolic changes in human macrophages differ from those in *S.* Typhimurium-infected murine macrophages. This provides insight into the limitations of the *S.* Typhimurium model for studying the pathogenesis of typhoid fever caused by *S.* Typhi. Furthermore, our results enhance the understanding of *Salmonella* pathogenesis and pathogen–host interaction at the metabolic level.

## 4. Materials and Methods

### 4.1. Bacterial Culture

The various *S.* Typhi strains and plasmids used in this study are listed in Appendix A. The Ty2 strain was used as the wild-type variant. The mutant strains were constructed using the previously described λ red recombinase system [39]. The primer sequences used for the generation and confirmation of the mutant are listed in Appendix A. The bacterial strains were grown in Luria-Bertani (LB) broth at 37 °C with shaking at 180 rpm or on LB plates. The LB medium was supplemented with the appropriate antibiotics at the following concentrations: ampicillin, 100 μg/mL; blasticidin S, 100 μg/mL; chloramphenicol, 25 μg/mL; kanamycin, 50 μg/mL; gentamicin, 20 or 100 μg/mL.

### 4.2. Cell Culture

Primary human peripheral blood CD14+ mononuclear cells were purchased from Stem Cell Technologies (Vancouver, BC, Canada). The Human THP-1 monocyte-like cell line (ATCC TIB-22) was purchased from the Shanghai Institute of Biochemistry and Cell Biology of the Chinese Academy of Sciences (Shanghai, China). The cells were cultured in RPMI-1640 medium supplemented with 10% fetal bovine serum (FBS) at 37 °C in 5% CO_2_. The CD14+ mononuclear cells were differentiated to macrophages by the administration of recombinant 1% macrophage colony-stimulating factor for 7 days before infection. The THP-1 cells were differentiated to macrophages by the administration of 50 ng/mL phorbol 12-myristate 13-acetate for 48 h before infection.

### 4.3. Lentiviral Vector Construction and Transfection

For the knockdown of *slc2a1*, a lentiviral shRNA vector targeting human *slc2a1* was constructed by Genomeditech (Shanghai, China). The recombinant vectors were prepared as lentivirus to transduce knockdown. THP-1 cells were incubated with the cell culture medium containing recombinant lentiviruses for 24–48 h. The culture medium was replaced with a medium containing 1 μg/mL puromycin for 6–9 days to establish a stable cell line.

### 4.4. Infection of Macrophages

*S.* Typhi strains were grown overnight in LB medium to obtain stationary phase bacteria [39]. The bacterial culture was diluted in RPMI-1640 containing 10% FBS and incubated at 37 °C for 20 min. Cells in 12-well plates were infected with bacteria at a multiplicity of infection of 10, and the cell plates were subsequently centrifuged at 800× *g* for 5 min. After 30 min of phagocytosis, the cells were washed 3 times with phosphate-buffered saline (PBS) and incubated in RPMI-1640 medium containing 100 μg/mL gentamicin for 1 h to kill any extracellular bacteria. The cells were then washed twice with PBS and transferred to RPMI-1640 medium containing 20 μg/mL gentamicin for the remaining period of infection. To calculate the intracellular bacterial replication, the infected cells were lysed with 1% Triton X-100 and serially diluted for plating on LB agar. Intracellular replication was calculated as fold change in CFUs between 2 h and 20 h post-infection.

### 4.5. Seahorse Analysis

The OCR and ECAR in human primary MDMs were determined using an XF24 Extracellular Flux Analyzer (Seahorse Bioscience, North Billerica, MA, USA). The XF glycolysis stress test kit, including D-glucose and 2-DG, and the XF cell mito stress test kit, including oligomycin, rotenone and antimycin A, were obtained from Seahorse Bioscience Inc. The cells were plated in an XF24 plate at 1 × 10^5^ cells per well and differentiated to macrophages. Before the experiment, cells were mock-infected or infected with *S.* Typhi for 8 h as described in Section 4.4. For the OCR analysis, the cells were washed 3 times and incubated without CO_2_ for 1 h at 37 °C in XF assay medium containing 2 mM L-glutamine, 10 mM glucose, and 1 mM pyruvate. For the ECAR analysis, the cells were treated with XF assay medium containing 1 mM L-glutamine. The ECAR measurements were calculated by subtracting non-glycolytic acidification values from the average post-glucose ECAR values. The basal OCR was calculated by subtracting nonmitochondrial respiration from the values obtained prior to oligomycin addition.

### 4.6. RNA Sequencing and Analysis

Total RNA was extracted from the cells after *S.* Typhi infection or mock-infection using an RNeasy Mini Kit (QIAGEN, Germantown, MD, USA). RNA sequencing and analysis was performed by Genewiz (Suzhou, China). Pass filter data in the FASTQ format were processed using Cutadapt (v1.9.1) [40]. Hisat2 (v2.0.1) was used for indexing of reference genome sequences and alignment to the human genome (*Homo sapiens*, GCA_000001405.28) [41]. The read numbers mapped to each gene were calculated using HTSeq (v0.6.1) [42]. Differential expression analysis was performed using the DESeq2 Bioconductor package [43]. All RNA sequencing datasets were deposited into the publicly available Sequence Read Archive (SRA) data repository (accession code: PRJNA721701; https://www.ncbi.nlm.nih.gov/bioproject/PRJNA721701, accessed on 14 September 2021).

### 4.7. Metabolomics Analysis

MDMs were mock-infected or infected with *S*. Typhi for 8 h, after which the cells were harvested and washed with pre-cooled saline to remove the medium. Subsequently, metabolites were extracted using a solvent (40% methanol, 40% acetonitrile, and 20% distilled H_2_O) containing 0.1% formic acid and incubated at −20 °C for 20 min. The extract was subsequently centrifuged for 10 min at 4 °C and 12,000× *g* to obtain the supernatant for metabolite analysis, which was conducted using ultra-high-performance liquid chromatography (Acquity; Waters, Milford, MA, USA) coupled with mass spectrometry (Q Exactive Hybrid Quadrupole-Orbitrap; Thermo Fisher Scientific, Waltham, MA, USA). Metabolites were separated, detected, and quantified as described previously [44]. Briefly, metabolites were separated on a Luna NH_2_ 3 μm, 2 mm × 100 mm column (Phenomenex, Torrance, CA, USA). The mobile phases comprised 20 mM ammonium acetate adjusted to pH 9.0 with ammonium hydroxide (A) and acetonitrile containing 0.1% formic acid (B). The mass spectrometer with a heated electrospray ionization source was operated in positive/negative modes. The key parameters were as follows: ionization voltage, +3.8 kV/−3.0 kV; sheath gas pressure, 35.0 arbitrary units; capillary temperature, 320 °C. The mass spectrometer was operated in full scan (70–1000 *m*/*z*, 70,000 resolution). Metabolomic data were processed with Xcalibur 4.0 (Thermo Fisher, Waltham, MA, USA). Metabolite identification was conducted using high-resolution mass and retention-time matching to authentic standards. Metabolite abundance was normalized to the cell number. Three biological replicates of each sample were analyzed.

### 4.8. Glucose Uptake Assay

Host cell glucose uptake was measured as described previously [45]. THP-1 macrophages were seeded into 96-well cell plates and mock-infected or infected with *S.* Typhi for 8 h. After washing 3 times with PBS, the cells were incubated at 37 °C in glucose-free RPMI-1640 containing 100 μg/mL 2-NBDG for 30 min; the fluorescence intensity of 2-NBDG (Excitation/Emission = 465 nm/520 nm) was measured using a Spark multilabel plate reader (Tecan, Männedorf, Switzerland).

### 4.9. Fluorescence Microscopy

THP-1 cells were grown on coverslips and infected with *S.* Typhi. To visualize the relationship between bacteria and host cell glucose transport, glucose-free medium with 100 μg/mL of 2-NBDG was added to the coverslips, which were subsequently incubated at 37 °C for 30 min. Cell monolayers were fixed in 4% paraformaldehyde and stained with 1 μg/mL Hoechst-33258 for 20 min. Subsequently, the cells were sealed with a fluorescence quencher and imaged using fluorescein isothiocyanate and mCherry absorption/emission settings in a laser scanning confocal microscope running Zen software (LSM 800; ZEISS, Oberkochen, Germany; https://www.zeiss.com.cn/microscopy/products/microscope-software/zen.html#inpagetabs-5, accessed on 14 September 2021).

### 4.10. Intracellular Glucose Measurement

*S.* Typhi-infected cells were lysed with 0.1% Triton X-100 for 10 min, and the glucose level was determined using the Amplex Red glucose/glucose oxidase assay kit (Invitrogen, Waltham, MA, USA). The opaque 96-well plate was measured on a Spark multilabel plate reader (Tecan, Männedorf, Switzerland) according to the instruction manual. Glucose concentrations were calculated using a standard curve.

### 4.11. Animal Experiments

Humanized NOD.*Cg-Prkdc^em1IDMO^Il2rg^em2IDMO^* mice (NOD-*Prkdc*^null^ *IL2Rγ*^null^, NPI) were purchased from BEIJING IDMO Co., Ltd. (Beijing, China). All experiments were conducted according to protocols approved by the Institutional Animal Care Committee at Nankai University (Tianjin, China). The mice were housed under pathogen-free conditions with a 14 h light/10 h dark cycle and fed ad libitum with mildly acidified water and standard chow.

Humanized mice were infected intraperitoneally with a 1:1 mix of *S.* Typhi wild-type or Δ*ptsG/manXYZ/glk* (1 × 10^4^ CFUs in total) and euthanized after 72 h; the livers and spleens were harvested and homogenized in PBS. The bacteria in tissue homogenate dilution were counted using a plate counting method. The competitive index of the wild-type and mutant was calculated as previously described [46].

### 4.12. Statistical Analysis

All experimental data were analyzed using Prism version 8.0.1 (GraphPad Software, San Diego, CA, USA; https://www.graphpad.com/scientific-software/prism/, accessed on 14 September 2021). Student’s unpaired *t*-test, one-way ANOVA, or the Mann–Whitney U-test was performed according to the requirements of the data. Statistical significance was set at *p* < 0.05. All the in vitro experiments were conducted in triplicate. The animal experiments were conducted in duplicate; the combined data of two experiments were used for statistical analysis.

## Figures and Tables

**Figure 1 ijms-22-10003-f001:**
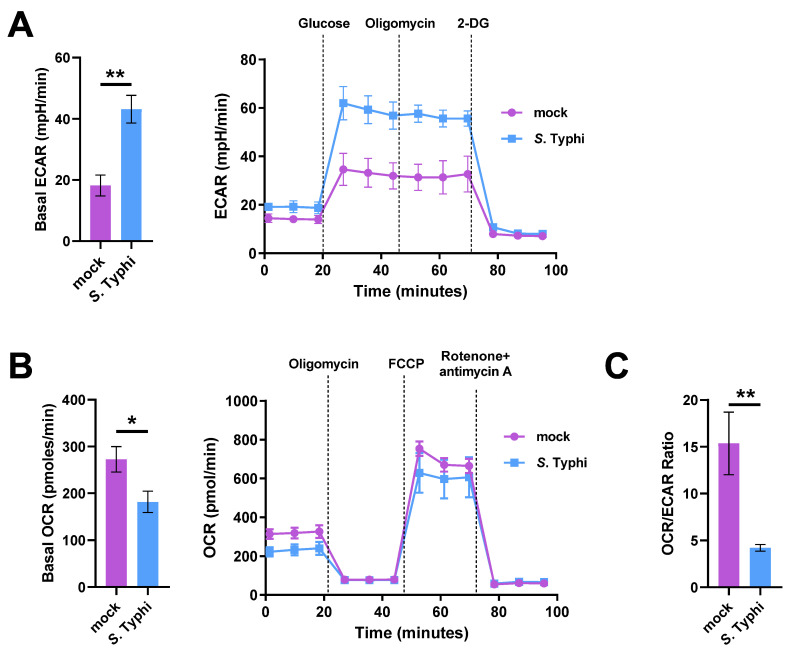
Analysis of glycolytic flux shows *Salmonella enterica* serovar Typhi (*S.* Typhi) infection increases glycolysis while decreasing oxidative phosphorylation (OXPHOS) in human macrophages. (**A**) Extracellular acidification rate (ECAR) of *S*. Typhi-infected human primary monocyte-derived macrophages (MDMs) vs. mock-infected MDMs (left), and real-time changes in the ECAR (right); 2-deoxy-d-glucose (2-DG). (**B**) Oxygen consumption rate (OCR) of *S*. Typhi-infected MDMs compared to mock-infected MDMs (left) and real-time changes in the OCR (right); carbonyl cyanide *p*-(tri-fluromethoxy)phenyl-hydrazone (FCCP). (**C**) OCR/ECAR ratio of mock-infected or *S*. Typhi-infected MDMs. Data were obtained from three independent experiments and analyzed using Student’s *t*-test. * *p* < 0.05; ** *p* < 0.01.

**Figure 2 ijms-22-10003-f002:**
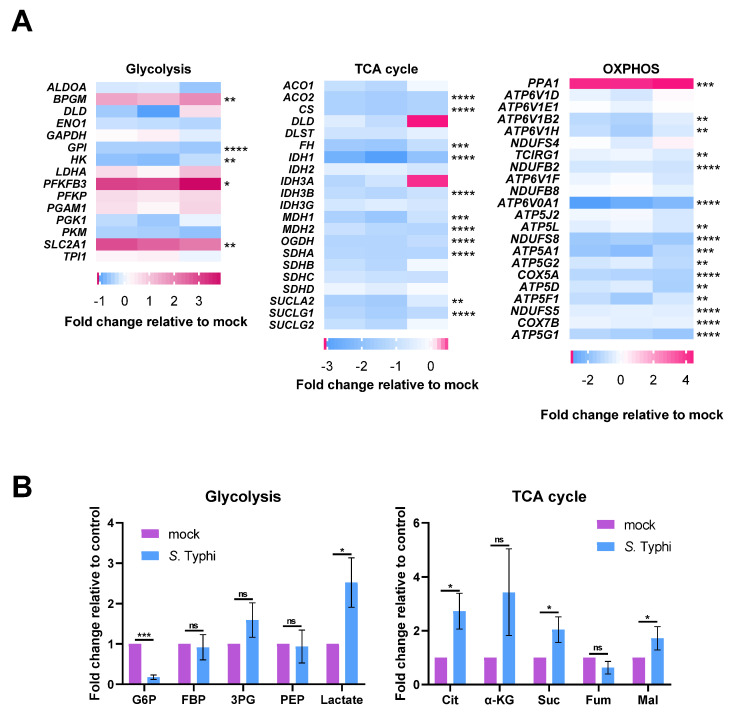
Transcriptomes and metabolomics analysis show *Salmonella enterica* serovar Typhi (*S.* Typhi) infection increases glycolysis while suppressing the tricarboxylic acid (TCA) cycle and oxidative phosphorylation (OXPHOS) in human macrophages. (**A**), Heat map of the gene expression of glycolytic metabolism (left), the TCA cycle (middle), and OXPHOS (right) in *S*. Typhi-infected MDMs vs. mock-infected MDMs. Fold changes of the relative gene expression levels are displayed in the heat maps, with pink representing higher and blue representing lower abundance. (**B**) Fold change in metabolites of glycolysis (left) or the TCA cycle (right) in *S*. Typhi-infected MDMs vs. mock-infected MDMs. Glucose 6-phosphate (G6P), fructose-1,6-diphosphate (FBP), 3-phosphoglycerate (3PG), phosphoenolpyruvate (PEP), citrate (Cit), α-ketoglutarate (α-KG), succinate (Suc), fumarate (Fum), malate (Mal). (**A**,**B**) Data were obtained from three independent experiments and analyzed using one-way ANOVA. * *p* < 0.05; ** *p* < 0.01; *** *p* < 0.001; **** *p* < 0.0001; ns, not significant.

**Figure 3 ijms-22-10003-f003:**
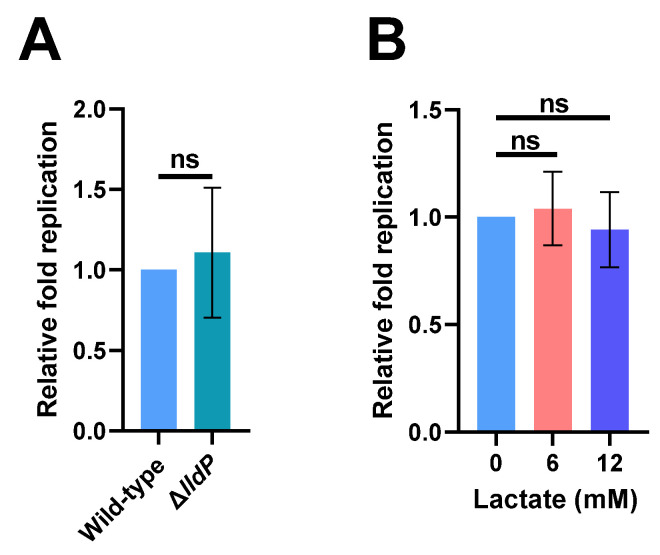
Increased lactate production in macrophages has no effect on *Salmonella enterica* serovar Typhi (*S.* Typhi) intracellular replication. (**A**) Replication of wild-type *S*. Typhi and Δ*lldP* mutant strain that cannot utilize lactate in THP-1 cells. (**B**) Replication of *S*. Typhi in 0, 6, or 12 mM lactate-treated THP-1 cells. (**A**,**B**) Data were obtained from three independent experiments and analyzed using Student’s *t*-test. ns, not significant.

**Figure 4 ijms-22-10003-f004:**
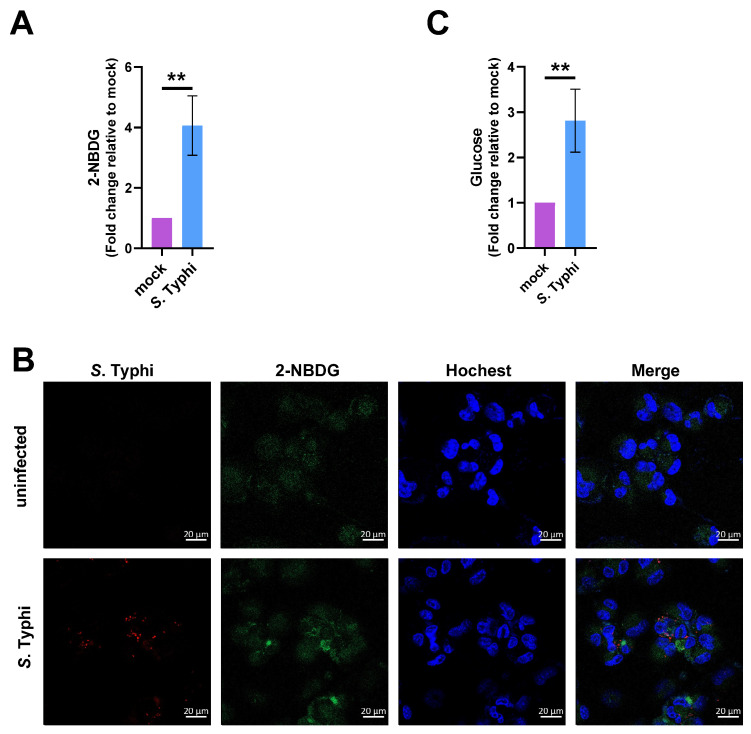
*Salmonella enterica* serovar Typhi (*S.* Typhi) infection increases the glucose level in human macrophages. (**A**) Glucose uptake assay in mock-infected or *S*. Typhi-infected THP-1 cells in glucose-depleted RPMI-1640 medium, using the fluorescent glucose analog, 2-(*N*-(7-nitrobenz-2-oxa-1,3-diazol-4-yl) amino)-2-deoxyglucose (2-NBDG). (**B**) Representative images from three independent experiments showing 2-NBDG distribution in mock-infected or *S*. Typhi-infected THP-1 cells. Red, *S*. Typhi; green, 2-NBDG; blue, nuclei. Scale bars represent 20 μm. (**C**) Glucose uptake in mock-infected and *S*. Typhi-infected THP-1 cells. (**A**,**C**) Data were obtained from three independent experiments and analyzed using Student’s *t*-test. ** *p* < 0.01.

**Figure 5 ijms-22-10003-f005:**
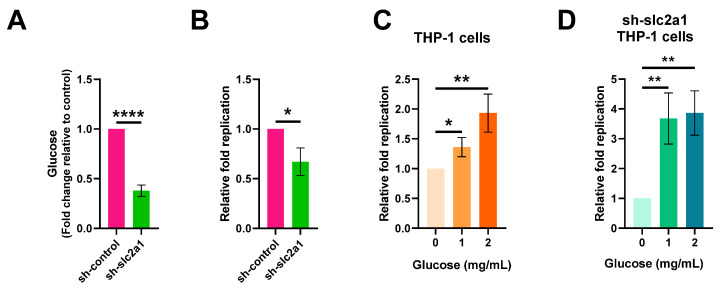
Increased macrophage glucose levels contribute to intracellular replication of *S. enterica* serovar Typhi (*S.* Typhi). sh-*slc2a1* represents treatment with lentiviral short hairpin RNA (shRNA) to reduce intracellular glucose levels in human macrophage cell line THP-1 (**A**) Glucose levels of sh-*slc2a1* and sh-control-treated THP-1 cells. (**B**) Replication of *S*. Typhi in sh-*slc2a1* or sh-control-treated THP-1 cells. (**C**) Replication of *S*. Typhi in the presence of 0 mg/mL, 1 mg/mL, or 2 mg/mL glucose, in THP-1 cells. (**D**) Replication of *S*. Typhi in the presence of 0 mg/mL, 1 mg/mL, or 2 mg/mL glucose, in sh-*slc2a1* treated THP-1 cells. (**A**–**D**) Data were obtained from three independent experiments and analyzed using Student’s *t*-test. * *p* < 0.05; ** *p* < 0.01; **** *p* < 0.0001.

**Figure 6 ijms-22-10003-f006:**
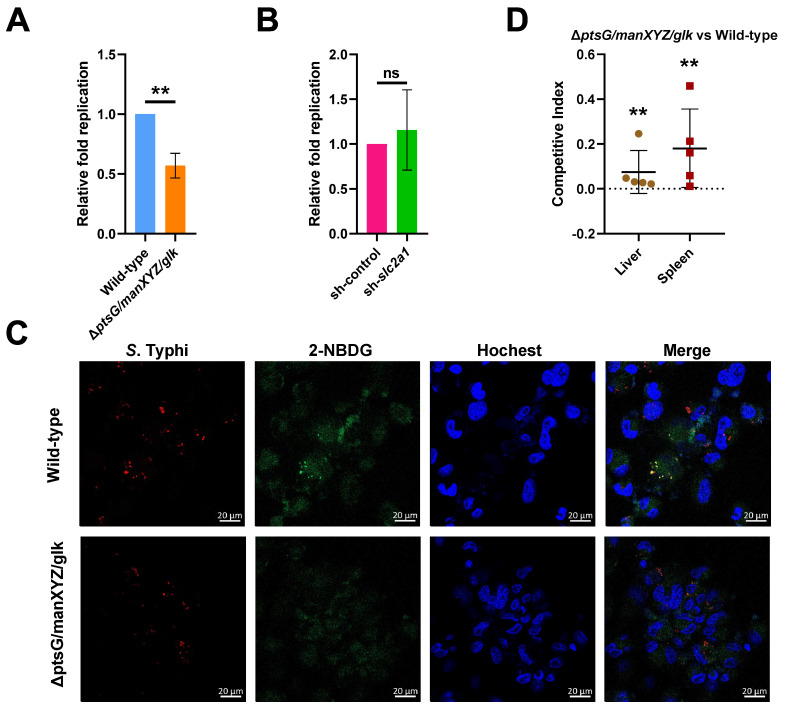
*Salmonella enterica* serovar Typhi (*S.* Typhi) uses host-derived glucose as a nutrient for intracellular replication and systemic infection. (**A**) Replication of wild-type *S*. Typhi and the Δ*ptsG/manXYZ/glk* mutant strain in THP-1 cells, and (**B**) replication of Δ*ptsG/manXYZ/glk* in sh-*slc2a1* or sh-control THP-1 cells. (**A**,**B**) The data were obtained from three independent experiments and analyzed using Student’s *t*-test. (**C**) Representative images showing the distribution of the fluorescent glucose analog, 2-(*N*-(7-nitrobenz-2-oxa-1,3-diazol-4-yl) amino)-2-deoxyglucose (2-NBDG) distribution in wild-type or Δ*ptsG/manXYZ/glk*-infected THP-1 cells. Red, bacteria; green, 2-NBDG; blue, nuclei. Scale bars represent 20 μm. The images are representative of three independent experiments. (**D**) Competitive index of Δ*ptsG/manXYZ/glk* vs. the wild-type in the liver and spleen of humanized mice. The data from two independent experiments were combined and analyzed using the Mann–Whitney U-test based on colony-forming unit counts of Δ*ptsG/manXYZ/glk* vs. wild-type. ** *p* < 0.01; ns, not significant.

## Data Availability

The RNA-seq data acquired in this study are available in Sequence Read Archive (SRA) data repository (accession code: PRJNA721701, https://www.ncbi.nlm.nih.gov/bioproject/PRJNA721701, accessed on 14 September 2021). Other data are presented within manuscript and Appendix A.

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
