# Peer review of "Salmonella enterica Serovar Typhi Induces Host Metabolic Reprogramming to Increase Glucose Availability for Intracellular Replication"

_ijms, 2021, doi:10.3390/ijms221810003_

Round 1

Reviewer 1 Report

Title – grammar for Salmonella Typhi – if authors are referring to the serotype typhi – they should spell that our properly – alternatively they can use standard nomenclature with lowercase subspecies enterica typhi or used serovar and then Typhi in the title.

Consistency with nomenclature in title – abstract and introduction needed

Line 35 – transmission occurrence via contaminated food or water. – include reference here

Line 47 – is SPI-7 equivalent S. typhi SPI-2 from typhimurium? If so, state such as it is unclear from text.

Line 61 – define Warburg effect – authors state Warburg-like metabolism without the needed introductory context.

Line 72 – this suggests that the metabolic effect and Warburg are not similar, but context would imply authors are suggesting similarity. Between S. typhi and S. typhimurium in terms of metabolic modulations. This needs a bit of clarity for the reader.

Line 78 – italicize in vitro and in vivo

Figure 1 – A – ECAR for the figure on the left – is this total from the timeline on right? The figure on the right needs elaboration – what is being measured with oligomycin and 2-DG in the experimental procedure? This is unclear.  – The same comments for part B as well.

figure 4 B - images need to have magnification included - authors state scale bar but is not clear in the image presented. 

Discussion - if authors could elaborate on the increased uptake of glucose and the increase in glycolysis to correlate with their description of the Warburg like phenotype with the decrease in hexokinase. 

Author Response

Thank you for your patient and thoughtful reading as well as the constructive comments about our manuscript. We have revised the manuscript based on your comments and suggestions. Our point-by-point responses to the specific comments are provided below.

SPECIFIC COMMENTS

Point 1: Title – grammar for Salmonella Typhi – if authors are referring to the serotype typhi – they should spell that our properly – alternatively they can use standard nomenclature with lowercase subspecies enterica typhi or used serovar and then Typhi in the title.

Consistency with nomenclature in title – abstract and introduction needed

Response 1: Thank you for the comment. The title has been changed to “Salmonella enterica serovar Typhi Induces Host Metabolic Reprogramming to Increase Glucose Availability for Intracellular Replication”. The nomenclature Salmonella enterica serovar Typhi” has now been used throughout the revised manuscript.

Point 2: Line 35 – transmission occurrence via contaminated food or water. – include reference here

Response 2: Done (line 37).

Point 3: Line 47 – is SPI-7 equivalent S. typhi SPI-2 from typhimurium? If so, state such as it is unclear from text.

Response 3: Sorry for the confusion. SPI-7 is not equivalent S. Typhi SPI-2 from S. Typhimurium.

SPI-2 is present in both S. Typhimurium and S. Typhi. The expression of SPI-2 gene is essential for S. Typhimurium replication inside murine macrophages and causing systemic infection in mice (Mol Microbiol, 1998, 30(1): 163–174), but are not required for S. Typhi replication in human macrophages and causing human systemic infection (Cell Host Microbe, 2019, 26(3): 426–434; Microbiology, 2010, 156: 3689–3698).

SPI-7, which encodes a Vi capsular antigen, is present in S. Typhi but absent from the genome of S. Typhimurium. It has been reported that the SPI-7-encoded Vi capsular antigen contributes to the systemic virulence of S. Typhi (Microbiology,2015, 13(4): 206–216; Cell Host Microbe, 2019, 26(3): 426–434).

To avoid confusion, we have added more information about SPI-2 and SPI-7 in the revised manuscript (lines 47, 50).

Point 4: Line 61 – define Warburg effect – authors state Warburg-like metabolism without the needed introductory context.

Response 4: Thank you for pointing this out. We have added more information for Warburg-like metabolism in the revised manuscript (lines 62–63).

Point 5: Line 72 – this suggests that the metabolic effect and Warburg are not similar, but context would imply authors are suggesting similarity. Between S. typhi and S. typhimurium in terms of metabolic modulations. This needs a bit of clarity for the reader.

Response 5: Thank you for the comment. Our previously work showed that S. Typhimurium infection also induces a Warburg-like metabolism in murine macrophages. Moreover, S. Typhimurium suppressed macrophage serine synthesis pathway via the type III secretion system (T3SS) effector SopE2 to promote the accumulation of glycolytic intermediates, especially 3PG, pyruvate and lactate in infected macrophages, for its intracellular replication (Nat Commun, 2021, 12(1): 879). We have now added the information in the revised manuscript (lines 73–75).

Point 6: Line 78 – italicize in vitro and in vivo

Response 6: Done (lines 15, 83, 276, 467).

Point 7: Figure 1 – A – ECAR for the figure on the left – is this total from the timeline on right? The figure on the right needs elaboration – what is being measured with oligomycin and 2-DG in the experimental procedure? This is unclear.  – The same comments for part B as well.

Response 7: Thank you for the comment. We showed basal rate of ECAR and OCR in the original Figure 1A (left) and 1B (left), respectively. The basal ECAR and basal OCR value indicate the rate of glycolysis and mitochondrial respiration, respectively. The basal ECAR values were obtained by subtracting the nonglycolytic acidification values from the three averaged measurements after glucose injection. The basal OCR values were obtained by subtracting the nonmitochondrial respiration values from the values prior to oligomycin addition.

To avoid confusion, we now used “Basal ECAR” and “Basal OCR” as Y axis titles in the revised Figure 1A (left) and 1B (left), respectively. 

Point 8: figure 4 B - images need to have magnification included - authors state scale bar but is not clear in the image presented.

Response 8: We have magnified the scale bar in the revised Figure 4B and the revised Figure 6C.

Point 9: Discussion - if authors could elaborate on the increased uptake of glucose and the increase in glycolysis to correlate with their description of the Warburg like phenotype with the decrease in hexokinase.

Response 9: Thank you for the comment. We have revised the Discussion section to indicate the relationship among “the increased uptake of glucose and glycolysis”, “Warburg like phenotype”, and “the decrease in hexokinase” (lines 308–315).

Reviewer 2 Report

The manuscript discloses interesting findings of host-pathogen interactions for common pathogen Salmonella enterica serovar typhi (S. Typhi). Integrated omics techniques (metabolomics and transcriptomics) as well as some other assays such as glucose uptake assay and fluorescence microscopy were employed to decipher the potential mechanism underlying S. Typhi pathogenesis. Overall, I do believe the manuscript is quite important to the field however a revision is required before publication.

-The metabolomics experiment dose not seem targeted metabolomics based on the description in the methodology it should be untargeted. Furthermore, the authors need to delineate the methodology section related to metabolomics as its ambiguous e.g. how many biological and technical replicates were used, data processing and statistical analysis need to be stated clearly too.

-Italicize [ in vivo and in vitro] as well as bacterial species throughout the manuscript.

-Line 92, add [s] to [Figure].

-Figure 2A, glycolysis panel the fold change scale needs to be similar to other pathways.

Author Response

GENERAL COMMENTS

The manuscript discloses interesting findings of host-pathogen interactions for common pathogen Salmonella enterica serovar typhi (S. Typhi). Integrated omics techniques (metabolomics and transcriptomics) as well as some other assays such as glucose uptake assay and fluorescence microscopy were employed to decipher the potential mechanism underlying S. Typhi pathogenesis. Overall, I do believe the manuscript is quite important to the field however a revision is required before publication.

Thank you for your patient and thoughtful reading as well as the constructive comments about our manuscript. We have revised the manuscript based on your comments and suggestions. Our point-by-point responses to the specific comments are provided below.

SPECIFIC COMMENTS

Point 1: The metabolomics experiment dose not seem targeted metabolomics based on the description in the methodology it should be untargeted. Furthermore, the authors need to delineate the methodology section related to metabolomics as its ambiguous e.g. how many biological and technical replicates were used, data processing and statistical analysis need to be stated clearly too.

Response 1: Thank you for the comment. The detailed method for metabolomics experiment and analysis is now described in the revised manuscript (lines 412–423).

Point 2: Italicize [in vivo and in vitro] as well as bacterial species throughout the manuscript.

Response 2: As suggested, “in vivo and in vitro” were italicized (lines 15, 83, 276, 467), and bacterial species were italicized throughout the revised manuscript.

Point 3: Line 92, add [s] to [Figure].

Response 3: Done (line 97).

Point 4: Figure 2A, glycolysis panel the fold change scale needs to be similar to other pathways.

Response 4: Done (revised Figure 2A).